# The Role of Workplace on Work Participation and Sick Leave after a Terrorist Attack: A Qualitative Study

**DOI:** 10.3390/ijerph18041920

**Published:** 2021-02-17

**Authors:** Trond Heir, Elise Hansen Stokke, Karina Pauline Tvenge

**Affiliations:** 1Norwegian Centre for Violence and Traumatic Stress Studies, N-0484 Oslo, Norway; 2Institute of Clinical Medicine, University of Oslo, N-0450 Oslo, Norway; 3Department of Psychology and Behavioural Sciences, Aarhus University, DK-8000 Aarhus, Denmark; elise.hansen.stokke@gmail.com (E.H.S.); karinatvenge@gmail.com (K.P.T.)

**Keywords:** workplace terror, work stress, resilience, disaster preparedness, leadership, qualitative research

## Abstract

Returning to work after large-scale traumatic events is desirable for employees, their organization, and society. The aim of the present study was to identify work-related factors that are perceived as important for work participation versus sick leave after a terrorist attack. We conducted in-depth interviews of 98 employees in the Norwegian governmental ministries that were the target of the 2011 Oslo bombing. Participants were randomly selected from 2519 employees who had responded to a web-based survey. We used a stratified sampling procedure to ensure inclusion of a wide range of experiences in terms of exposure and stress reactions. Participants were asked what, if any, factors contributed to work participation or sick leave, and which factors made a difference in how quickly people on sick leave returned to work. Thematic analyses provided three themes that stimulate work participation and prevent sick leave: supportive management, the ability of a leader to accept individual needs and help people cope with stress; sense of cohesion, feelings of being close, caring for each other, and working well together; and working as a coping strategy, basic assumptions that it is best to stick to work and familiar routines, or a strong belief in one’s ability to master. A fourth theme, high demands and lack of acceptance, included experiences that promoted an absence from work, such as too much business as usual, management’s lack of priorities for which tasks could be left out, or a lack of recognition of individual needs. The findings point to key factors that workers perceive as important for work participation in the aftermath of a disaster. We suggest that health and productivity benefits can be achieved by organizing work and the work environment in line with these experiences.

## 1. Introduction

A large body of literature has documented that traumatic events, such as disasters or terrorist attacks, affect public health and result in posttraumatic stress, anxiety, depression, and comorbid mental or physical disorders [1,2,3]. The resulting health adversities may affect the workability of workers and contribute to sick leave. [4,5]. Thus, work participation and sick leave after large-scale traumatic events is an important issue for research, and the role of the work environment warrants attention [6].

Associations between the work environment and sick leave under normally stable conditions have been investigated in several studies. Systematic reviews have stated that work overload, lack of control over work pace and decision making, lack of support, and unclear management or work roles are associated with increased sick leave [7]. In addition, a successful return to work after sick leave is predicted by factors related to job stressors and organizational stress [8]. However, the role of the working environment in enabling employees to stay in work in the aftermath of a traumatic event has rarely been investigated. Following the 2011 Oslo bombing, control over decisions at work, support from superiors, and support from co-workers reduced the odds of sick leave, whereas role conflicts increased the odds [9]. For employees who met the symptom criteria of posttraumatic stress disorder (PTSD), predictability and control over decisions at work reduced the probability of sick leave [10]. To the best of our knowledge, no post-trauma research has been conducted using qualitative in-depth interviews. 

The few studies that have investigated work-related factors and sick leave in the aftermath of traumatic events have used the general Nordic questionnaire for psychological and social factors at work (QPSNordic) which was developed for working life research in general [11]. Therefore, they have some important limitations in their ability to identify conditions that may be specific to traumatic events. To recognize new or specific experiences for disaster victims, research should be conducted in the form of in-depth interviews of the working staff. A qualitative approach may reveal more nuance in the employees’ experiences and provide a more realistic knowledge base for later application.

The purpose of the present study was to identify factors at work and in the work environment that workers perceived to be important for work participation, sick leave, or returning to work after absenteeism in the aftermath of a workplace terrorist attack. 

## 2. Materials and Methods

### 2.1. Participants

We used in-depth interviews of ministerial employees who were the target of the 2011 Oslo bombing, in which a car bomb explosion in the executive government quarter shattered government buildings, killing 8 people and injuring 209 more. We selected 100 participants from 2519 employees who had completed a web-based three-wave survey 10, 22, or 24 months after the terrorist attack [12]. The original survey participants were sampled on a voluntary basis following a request to all employees in 14 of 17 Norwegian governmental ministries. Respondents comprised 70.4% of individuals who were employed on the day of the attack.

To ensure that the study included a wide range of participants in terms of exposure and stress reactions, we used a stratified sample based on two criteria: whether survey participants were present in the government district at the time of the explosion (directly exposed) or not (indirectly exposed), and whether they at any wave in the follow-up survey met the symptom criteria for a diagnosis of PTSD. Detailed information on the actual criteria was published previously [12]. We randomly selected employees in the following categories until we had 25 consenting participants in each: directly exposed PTSD cases, indirectly exposed PTSD cases, directly exposed non-PTSD cases, and indirectly exposed non-PTSD cases. The high number of participants was chosen to ensure experience across departments, subdivisions, positions, and education. We contacted a total of 113 employees, 9 of whom we failed to reach, and 4 who refused to participate in the interview.

During the research process, one individual dropped out due to technical tape failure, and another withdrew for personal reasons. Thus, 24 or 25 individuals in each category were subject to analysis, a total of 98 participants (Table 1).

### 2.2. Ethics

All participants provided written consent to participate, the tape recording of interviews, and the analysis and publication of results. The interviews handled sensitive topics, and safeguarding the participants was prioritized. The study was approved by the Norwegian Regional Committee for Medical and Health Research Ethics (reference no. 2011/1577 REK south-east D).

### 2.3. Procedure

The interviews were conducted between April and September 2015 by two trained interviewers in a sheltered environment at or near the participants’ workplace. The interviewers followed a semi-structured interview guide. The interview included questions about experiences with the terrorist attack, work requirements, coping strategies, and sick leave. The duration of the interview was approximately 1.5 h per person.

For the present study, the following questions were particularly relevant: Have you been on sick leave since 22 July 2011, and if so why? What, if any, work-related factors do you think contributed to your sick leave? Or what factors do you think contributed to the absence of sick leave? If you were on sick leave, what, if any, factors do you think made a difference in how quickly you returned to work?

All interviews were conducted in Norwegian, recorded, and transcribed in Norwegian. The extracts selected for this publication were translated into English by a professional translator.

### 2.4. Thematic Analysis

We used thematic analysis with an inductive approach [13]. The two authors responsible for the analysis (EHS and KPT) had not previously conducted work environment research, and they approached the data material with a minimum of their own interests or theories that could have influenced the results or the choice of quotes in a desired direction. The analysis followed the six-step process of Braun and Clark [13]. Topics in the data were discussed amongst the authors until an agreement was reached. A thorough description of this process, including coding and identification of underlying themes, can be found in a Appendix A.

We used the software package NVivo [14] to classify, sort, and arrange the information. We present the main themes separately below. The themes are illustrated using quotes from the interviews. Due to the high number of participants, we have selected quotes that can be considered representative.

## 3. Results

After performing the thematic analysis, we ended up with four main themes: supportive management, sense of cohesion, working as a coping strategy, and high demands and lack of acceptance.

### 3.1. Supportive Management

The employees experienced that several management characteristics were important factors in their ability to work at full capacity following the terrorist attack, especially their managers’ capacity to accept and help their staff cope with stress. Having a supportive manager was highlighted as being important by 37 of the study participants. Of these participants, 29 had been directly exposed to the bomb blast and 20 had had symptoms consistent with a diagnosis of PTSD.

The employees experienced that it was important that the management were understanding and had confidence in them. Many participants were of the opinion that the management recognized stress reactions and accepted the employees’ reduced capacity to work effectively. This appeared to have motivated the employees and helped promote their capacity to work:


*It was [important] that they [the managers] showed compassion and empathy, and in general gave us a lot of slack when it came to day-to-day working. They expressed that they were fully aware that not everyone was feeling up to the task. In retrospect, it turned out that people did about as much work as they had done before, when they had only calmed down for a short period. ...I think the fact that it turned out as well as it did was because we gained the [managers’] confidence.*
(Female in her 60s)

The participants appreciated that all the managers were part of a general leadership culture that emphasized adjustment, support, and care, and that there was acceptance that different individuals had different needs:


*It was very important that the system, in other words my own employer, the ministry, but also the entire ministry community, with everything from occupational health services to the support network, had open arms and were so generous that we only had to say if we needed something. ‘If you need something, just say so. It does not matter what it costs, we will support you all the way.’ They also accepted that people’s individual needs were very different.*
(Female in her 50s)

The participants perceived it as positive that the management cared for everyone and did not discriminate among the employees. That positivity contributed to their confidence in the organization and an expectation that everyone contributed to the best of their ability:


*Those who worked to create the conditions so that we could do our job, they gave it their all and made a huge effort... So, we had a good feeling that if something could be done it would be done, and that we were all in the same boat. There was no discrimination. ...I think it [the situation] was very good. There was a good driving force in the organization ... and everyone contributed as much as they could.*
(Male in his 30s)

Self-determination and control over one’s own work was considered important by many of the employees. Of the 37 participants who described support from the managers as important for their capacity to work, 21 mentioned the facilitation of autonomy, especially the acceptance of an employee being able to determine their own drive and work pace:


*For me, it has been good to be allowed to slow down, at least sometimes, whether that means simply working from home or such as now, when I can slow down [my work pace] for a few weeks and let my shoulders sink a little ... Even if one is sometimes absent for a short time, it is important that one is still allowed to have interesting tasks.*
(Female in her 40s)

Although not everyone needed adjustments in their working conditions, merely having the opportunity to do so was important. It contributed to their sense of self-determination, which in turn helped prevent sick leave:


*I think the way we handled it had a preventative effect for many and possibly for me too. ...Just the knowledge that one could stay home if one needed to, was good.*
(Female in her 40s)

#### Discussion: Supportive Management

The benefit of supportive management fits well with several studies showing that leadership is associated with the health and well-being of workers in general [15,16,17], or that lack of leader support increases the probability of sick leave [7].

The benefit of supportive management also agrees with quantitative results from a larger sample of our study cohort [9,18]. However, the present findings clarify the actual employee experiences, especially related to the peculiar circumstances following a terrorist attack. The workplace is usually considered a safe place where people spend much of their lives, trusting the organization. A terrorist event strikes people when they are completely unprepared and shatter their basic assumptions of the world being a safe place [19]. When the workplace is the scene of the event, people may expect their managers to take responsibility for their well-being [20].

In this context, leader support appeared to increase the sense of security, increase the confidence and belief in people’s own coping skills, underpin information and guidance on how to deal with the peculiar work situation, recognize individual adaptations and needs, and facilitate coping with stress-related challenges, all of which were considered to protect against sick leave. The thematic analysis included more themes under supportive management than is usual in quantitative studies of working life. For example, the respondents mostly assigned responsibility for facilitating autonomy and clarity in roles to the management. These are often seen as independent themes that are positively associated with health or work participation, even within our study cohort [9,10,18].

The emphasis on self-determination and control over one’s own work is consistent with the literature on general working life, health, and well-being [21]. However, being able to decide one’s own progress and work pace can be even more important after expo-sure to a terrorist attack. Post-traumatic stress reactions, such as difficulties concentrating or emotional or relational problems, can hinder normal work progress and necessitate in-dividual day-to-day adjustments. The alternative to such adaptations could be sick leave.

### 3.2. Sense of Cohesion

We identified 61 employees who believed that a sense of cohesion in their workplace had a significant impact on their capacity to work. Of these employees, 28 had been directly exposed to the bomb blast and 24 had had symptoms that were consistent with a diagnosis of PTSD. Many employees experienced being part of a community in which there was high commitment. Despite practical challenges, such as having to move to new premises, reduced staffing levels, and lack of space and technology supplies, a consensus was found among their co-workers that they wanted to complete their assignments:


*We were very much characterized by a sense of unity. We were, of course, given temporary premises...and we had to be three or four people sharing an office, and we had to reconnect our PCs...we had been given some laptops... We had to make everything work and it took a few days before things fell into place. We knew that we had to prepare a state budget and that there were short deadlines. In fact, the atmosphere was quite good despite our circumstances. We all pulled together and were determined that we had to work a little more than usual, so we continued to stay on a little later in the afternoons to get the work done. ...There was a mood of rolling up one’s sleeves and so we kept going. ...It was ‘now we have to get the job done and finish the task.’*
(Male in his 30s)

Thus, it was evident that the employees were inspired by the strong sense of unity, as well as by the experience of overcoming the many challenges that arose because of the changes in their working conditions. It made sense for the employees to work well together. Several participants described how the sense of unity gave them positive energy and joy:


*Although it might seem like ‘swearing in church’, I thought it was great fun to get to that build-up phase, having that sense of unity. Despite the atrocity behind the situation, it was also incredibly good. It was a great time. People gave their all and did not care about anything else. It was a case of giving priority to the community and rebuilding it. So, despite the situation, it was fantastic. I am extremely glad I was part of that. ...Simply making things happen, making them flow smoothly, despite everything else. It was a lot of fun. It may be wrong to say ‘fun’, but it gave one a boost. ...The sense of unity was very good. ...Also, people did things they had never done before. They worked around the clock. There was a lot of laughter. ...Yes, despite the tragedy, there was something very positive, our memory of working together.*
(Female in her 50s)

The attitude of the management was important, particularly the fact that the managers were on the same teams as their staff and called for everyone to be united in their efforts. The experience of being stronger together had a positive effect on motivation and belief in mastery:


*What I have thought of as important for our workplace was that the Minister was very clear that we should achieve this [mastery]. We should get it together. We should not give up. ...I think the thought of not being a victim, but that we were strong enough, was very important. I think that set a standard for people. Everyone was looking for something, and it was crucial that the Minister said what he did, and that the management followed it up. It was not as though we were all on sick leave and that the best thing was to stay at home. No, everyone was expected to show up for work.*
(Female in her 40s)

Being well taken care of by work colleagues had a motivating effect on the employees’ capacity for work. Of the 61 participants who experienced unity being important for their work capacity, 36 emphasized the support given by their co-workers. For several of the employees who were on sick leave after the terrorist incident, their co-workers’ support was a significant motivating factor in their ability to return to work as soon as possible:


*Although I was on sick leave, I sat down here in the corner of the café. ...Then my good colleagues joined me for coffee. They took turns coming. ...I have been very lucky to have some close colleagues who have been there for me. For me, it felt like a safety net. ...I think that in a way they were invaluable. That was a time when I really needed to keep in touch with my colleagues.*
(Male in his 30s)

Contact with work colleagues during sick leave resulted in feeling that someone cared. That experience, together with receiving information about what was happening in the workplace, helped motivate the employees and make it easier to return to work: 


*Yes, I had very good support from the whole Ministry office when I was sick, right from the first few days and throughout the whole period I was in the hospital. One manager was incredibly supportive and came to visit in person and arranged for me to be visited by my work colleagues every week, almost in rotation. I think just about everyone in the office stopped by. ...So, I was continually kept informed about cases and what was happening. One feels...that it is a very good workplace one really wants to return to. There are people who care, and that means a lot. ...So, then when one is motivated, one just does it [returns].*
(Male in his 30s)

#### Discussion: Sense of Cohesion

The importance of cohesion was the most robust finding in our analysis. Most employees indicated that a sense of cohesion had significantly impacted their work participation, particularly the experience of being part of a community, working towards common goals, and having the support of colleagues. Historically, the concept of cohesion has occupied a key position in macro-sociology [22], as well as social psychology [23]. Our findings confirm the centrality of cohesion as a mediator of motivation [24], productivity [25], and even resilience after trauma [26]. Fostering connection quickly after mass trauma is considered critical for recovery [27,28], and there is broad consensus that promoting connectedness is a general aim of intervention strategies in the aftermath of disasters or terrorist events [29].

In ministerial work units where there had been good cohesion and a good working environment prior to the event, these characteristics were sometimes strengthened after the event. This was expressed in the employees’ descriptions of the spirit of working for common interests and the accompanying increase in driving force. The findings suggest that stress experiences and coping strategies may have collective qualities. Culture not only seems to moderate the appraisal of stress, but also contains collective coping responses, which seem to have an origin either in the organizational environment or the community [30]. Other authors have emphasized that resilience is not so much an individual construction as it is a quality of the environment and its ability to facilitate recovery [26], and that building resilience is a responsibility of the organization as a social system [31].

The employees’ assessment that the support from workmates promoted their work ability is in line with research from the working life in general [32,33]. The situation that arose after the terror attack appeared to have created an altruistic community that mobilized coworkers to provide social support beyond what was usual. This resulted in sharing a group identity, gaining emotional contact, and providing information or practical help, some of which arose spontaneously and some that was organized.

### 3.3. Working as a Coping Strategy

Many employees made a common assumption that it was best for them to go to work. They thought it was good to maintain familiar routines. For example, it was easier to deal with their reactions after the terrorist event by being in their usual work environment rather than staying at home. We identified 37 participants who stated that work was a form of coping strategy, 21 of whom had been directly exposed to the bomb blast and 20 who had had symptoms consistent with a diagnosis of PTSD:


*For me, just being at work and standing at work was precisely what was decisive for me doing so well [in the aftermath]. ...I did not need to take time off to sit at home and ponder. For me, that was neither attractive nor constructive. I knew that if one was alone in quietness, then one would only have oneself and one’s head to relate to. It is not certain that would be particularly positive. So, you know that what... [you must do is to get] out into the world.*
(Female in her 50s)

Being more prone to having painful thoughts while on sick leave was a common theme among the participants’ responses. Some participants thought it had been good for them to be able to focus on work-related tasks, whereas some merely needed to spend time with their work colleagues:


*It was never an option for me to consider being on sick leave...for me it is probably not good therapy to sit at home and look at the wall, and ponder, ponder, and ponder. Rather, it is better for me to be at work and in a context where I can keep my mind on professional matters, and where I can have social interactions with colleagues.*
(Male in his 30s)

Assumptions that it was better to be at work also motivated those who were on sick leave to return to work as soon as possible. Some participants talked about having a sense of normalization or to returning to everyday life as though “normal” or “everyday life” were preferable:


*Yes, because I was very aware that it was important for me to come back. ...So, I had to sit there and watch, and in that way try to experience a little normality again...I know it is not good to sit at home alone and think bad thoughts.*
(Female in her 50s)

Some employees expressed that they were pleased that a manager had encouraged them to return to work. They mentioned that they had been motivated by managers’ arguments that it was good “to have something to go to” or “to be with work colleagues”:


*My manager said, ‘I think it’s good for all of us to start working, as then you have something to go to and at least you will be together.’ When I think about that in retrospect, it was very wise. ...It felt very meaningful. I think it would have been harder if I had stayed at home. So, part of the reason why it was so good afterwards [i.e., after the bomb explosion] is that it was obvious to my manager, and then it became obvious to me, too. ‘Yes, of course we will start again.’*
(Female in her 40s)

Most participants who used work as a form of coping strategy also had a strong belief in self-mastery. They were reasonably confident that they would be able to complete whatever work they had started before the terrorist incident:


*I think I have learned that I can handle most things. I think that confirms... [that] I am quite a persistent person who does not give up my goals. ...At least, I think there is nothing I cannot do if I really want to. I think I will persevere at work, and even though sometimes I have gone home from work with a feeling of hatred, I think there is nothing I cannot cope with.*
(Female in her 20s)

Several participants emphasized that it was unacceptable to interrupt their work tasks. Moreover, it would conflict with their self-perception or identity:


*I think that is an old childhood lesson. I thought a lot about it then, that my father always said, ‘Whatever you have started, you must finish.’*
(Female in her 40s)

#### Discussion: Working as a Coping Strategy

Coping strategies based on a strong belief in the benefits of working differed from other themes in that they included the individual’s personality, motivation, previous experiences, or values to a greater extent. They also included attitudes that were communicated by leaders, but such input seemed to have the greatest impact if it fit into the individual’s basic values. In contrast, leaders with strong recommendations to maintain daily routines were in danger of appearing to lack the ability to meet individual needs.

Although the employee’s basic values appeared to be more important for work participation than the manager’s advice that it would be beneficial, some statements illustrated the effectiveness of the manager leading by example. This is in line with social learning theory, suggesting that the employees adopt the leader’s attitudes and behaviors through the leader acting as a role model [34].

The belief in the positive aspects of work is supported in the literature, which states that work is usually a source of health and mental health benefits [35]. For most people, the workplace serves as an essential social context that provides routine, purpose, and social resources in one’s life. Absence from work can result in the loss of resources that are important for recovery after traumatic events. For example, access to information and support can be significantly reduced. After the September 11, 2001 terror attack, more than twice as many of those with persistent distress accessed health information at work rather from a health professional, and more than three times as many sought information and counseling at work rather than from a community-based provider [36].

Returning to work after a workplace disaster will provide re-exposure, which is effective therapy for post-traumatic stress, including re-experiences that are easily triggered by reminders at the workplace [37,38]. Participation in work and daily routines are also important measures to restore feelings of security, trust, and organizational cohesion [39].

Even among those most affected, several believed in the benefit of working. This was often linked to a strong belief in one’s ability to cope with challenges, which the literature refers to as self-efficacy. The self-efficacy concept is usually tied less to specific tasks or situations but reflects a person’s ability to feel confident and rely on his or her own efforts to cope with the challenges of life in general [40]. Self-efficacy may promote recovery from stress reactions by using more active and adaptive coping strategies [41], including the desire to stay at work.

### 3.4. High Work Demands and Lack of Acceptance of Individual Needs

We identified 39 participants who described high demands for productivity or lack of acceptance of individual needs as inhibitive of work participation. Of these participants, 20 had been directly exposed to the bomb blast and 27 had had symptoms consistent with PTSD. Some participants experienced that the work demands were much the same as they had been before the terrorist incident. Although it was accepted that each employee could pay more attention to their own needs, some employees experienced that managers did not make active decisions about which work tasks could be given lower priority:


*It was ‘business as usual’. ...Yes, there was a lot of ‘Yes, now we must take care of each other, and we have to stop in the hallway and ask, “How are you?”’ However, at the same time it was a case of keep going. There was no reduction in the level of ambition, nor was there any clear prioritization. It was rather a case of ‘Just take a walk around the building if you feel sad...’ However, because no one gave less priority to their tasks, there was no time to take a short break.*
(Male in his 40s)

Some participants even experienced that the work demands became greater after the incident and that they had to take on more work than usual, either due to colleagues being on sick leave or a loss of infrastructure and office supplies, or because moving to new premises had made their work more demanding. The working conditions in the new premises were sometimes perceived as far from optimal and, according to the participants, the provision of care for employees’ well-being was given low priority:


*There was no such thing as ‘there may be some repercussions and that it may be someone thought that was rough’ or something like that. ‘No one has been injured [due to the blast]. We are working on the state budget, which must be finished, and we will return to work as soon as possible, even if we have to sweep the shards of glass ourselves.’ ...I felt I was standing at a crossroads. Either one pulls oneself together...and maybe brushes, sweeps glass away and writes the state budget and then get well through doing that…or one will fall by the wayside, which I did.*
(Male in his 50s)

Lack of recognition and high work demands were often presented as two sides of the same coin. For example, when the employees did not experience sufficient acceptance that there was a need for lower expectations about workload or work pace:


*The work environment, and especially the immediate line manager, did not understand that we could not just push through the paperwork. People were tired and they were traumatized. ... [The managers] simply rejected their wishes and views as if they were not important, [saying] ‘Now you just have to pull yourself together.’ I did it as best I could. She [the boss] had a way of accepting what came up, which was just completely...it just is not possible to behave like that. ...The pace of work went faster and faster. I worked increasingly more quickly. More and more. Some just sat there, completely paralyzed, unable to do anything, including even some who had not been there [i.e., at work during the explosion]. My way of solving what I found difficult was to work harder, but that did not go well. It only got worse and worse.*
(Female in her 40s)

According to the participants, some changes to processes had been initiated to either facilitate or increase the efficiency of the work. They might have perceived the latter as demanding, especially because the need for them to become acquainted with new digital solutions or forms of collaboration had come at the same time as they had to face other challenges. As shown by the following example, the changes in process became an additional burden that contributed to staff taking sick leave:


*Among other things, there was the fact that changes to processes were rolled out. It was very much about digital solutions and forms of collaboration that so far have been OK and good initiatives in themselves, but I think it was slightly...almost a little untimely. [Laughs] For me, who was there [at work during the explosion], I just wanted to be allowed to handle what tasks we were assigned on a daily basis... Those [changes] were large additional burdens for everyone, also for those who were not there [at work during the explosion]. One did not need the extra load. ...I think it contributed to extra stress.*
(Female in her 30s)

Some participants experienced less acceptance of employees talking about the incident after a while. In addition, the signs of higher work demand came too soon:


*However, I think in the beginning one should just talk about it [the bombing] and talk about it. One could say that when September came, then management wanted us to just work. One was supposed to not care... What was that about? So, when one was sitting in the common area where we can talk together, I heard, ‘What are you sitting here for?’ I think the management treated it [the situation] completely different to what, for example, the psychologist has recommended.*
(Female in her 50s)

The interviews also revealed that the employees differed in how they had been impacted by the terrorist incident. For those who needed to put the incident behind them, it became important to draw attention to other topics, but that risked consequences for those who felt the need to talk more about what had happened:


*A little lid was put on it and we were not supposed to [open it]. We sat in an open office landscape, where we should not talk about it [the bomb explosion] and...I can understand that, because the purpose was to take account of those who did not want to talk about it. But maybe it [the explosion] was a little too silenced. So, some of us came together for lunch or for a cup of tea or met in a room just so that we could sit and talk.*
(Female in her 40s)

Some participants experienced a form of hierarchy developing in the workplace according to who deserved the most understanding from colleagues and management. A significant distinction was made between those who had been at work during the terrorist incident and those who had not been at work. This became the basis for a form of group identity, allowing comparisons:


*Those who had been at work had a reason to grieve in a way, but for us who had just lost our workplace, it was not the same. ...That was experienced as somewhat unfair. It may sound strange to say it, but ‘See us, too. We are here. We have feelings. This is our job, too. We have also lost a colleague, even though we were not at work that day.’ So, there was a bit of a hierarchy in who could grieve and who could not. ...No one has the exclusive right to grief. This could perhaps have been made a little clearer in our ministry.*
(Female in her 40s)

In contrast, according to other participants, the management seemed very conscious of the fact that they should not discriminate between employees. Such discrimination could be at the expense of providing care for someone who felt they needed it more than others:


*In general, I think perhaps there was not that much care really. No, [on second thoughts] I do not think that. Perhaps it was also the fact that the management was reluctant to establish such a differentiation, ‘Those who were there, and those who were not there,’ such that the latter would experience that they were overlooked. I think they [the management] were very worried about that risk.*
(Female in her 50s)

#### Discussion: High Demands and Lack of Acceptance

Regardless of the terrorist event, many tasks had to be performed as usual in the ministries. For example, the basis for important political decisions had to be documented continuously, and it was imperative to prepare an annual state budget every autumn. The fact that more employees than usual were on sick leave in the first two years after the attack [5] further increased the work responsibilities of those present.

Sick leave due to experience with high demand and lack of acceptance of individual needs corresponds with research from working life in general, in which work overload and pressure increase the probability of sick leave [7]. Reduced work capacity due to post-traumatic stress, such as concentration or adaptation problems, may further increase the risk of overload. For some participants, adjustments to work ambitions were not sufficient, and a lack of facilitation significantly contributed to their sick leave. Furthermore, some experienced that the period of adaptation was too short and that business as usual took over before they were mentally restituted.

The balance between different needs may have been resolved differently in subdivisions of the ministries or, as some of the interviews suggested, reality was evaluated differently by different workers. Though some employees perceived the management as demanding, others praised the same leaders for their recognition of mental distress and individual needs. Opposite needs were expressed about the need for care, the preoccupation with who deserved care the most, and how long it was appropriate to focus on the terrorist event.

Experiences with high demands and lack of acceptance were in no way reserved for people who were directly exposed to the terror or had the highest levels of post-traumatic stress. The reason for the different attitudes and needs may have more to do with pre-event factors, such as personality, previous experiences, and position or role in the organization, than with the severity of exposure to the terrorist event itself. The manager’s attitudes and the relationship with the manager prior to the event may also be important.

## 4. Strengths and Limitations

This study is based on the employees’ subjective assessments of what contributed to work participation, making it vulnerable to attribution bias. In addition, the participants provided retrospective assessments, which allows for recall bias. Nevertheless, there is little reason to doubt that the topics raised by the participants had a significant impact on their health and well-being.

The quotes were selected to illustrate the topics they are intended to represent. The underlying choices raise questions about validity and generalizability. The fact that the analyses were performed separately by two researchers with a minimum of preconceived notions increases the validity. In addition, the large number of participants strengthened the opportunity to uncover some general patterns.

Several of the employees needed more support than the manager or colleagues could provide. This could affect how the employee perceived their management and the work environment. For example, employees with high stress reactions may perceive their manager to be less supportive [42].

Many leaders were themselves affected by the attack, but we have no data on how this affected their leadership. In addition, many leaders received guidance on how to lead people who have been victims of a terrorist attack, but we do not have data on how this affected their leadership. The participation of more women than men may have influenced the results. Female employees had more psychological distress after the incident [18], twice as high PTSD prevalence [12] and twice as high sickness absence as men [5]. No comparisons between genders have been made in this paper, and the direction of any gender bias is unknown.

The participants had, on average, higher education than the general population, which raises questions about the generalizability. Most of the employees had well-paid jobs and had built-up resilience in the face of stress and ever-changing working conditions. However, there were also participants with lower education levels, such as cleaning workers, kitchen staff, and security guards.

## 5. Implications

Over the past two decades, workplaces around the world have been increasingly exposed to natural and man-made disasters that destroy businesses and disrupt productivity, resulting in economic, social, and health consequences [39]. To promote mental health and sustain economic, social, and human capital, it is essential that interventions for preparedness, response, and recovery from workplace disasters occur in occupational settings.

Post-disaster interventions should be planned before an event occurs. The basis for management in future crises is found in today’s management. Surveys of the working environment in the years before and after the 2011 Oslo bombing have shown that the perception of leadership was remarkably stable from pre- to post-disaster [42]. Nevertheless, it may be appropriate to initiate leadership training in the aftermath of traumatic events to emphasize the importance of the leader accepting the individual needs of subordinates and helping them cope with stress [39]. Managers should have adequate knowledge of expected stress reactions, their nature and course, and the corresponding reduction in the capacity to work.

Post-traumatic approaches must often deal with the fact that no single solution suits everyone. Compromises or individual adjustments are needed, such as how appropriate it is to dwell on the past. In some ministries, the step was taken to mark lunch tables in one of two ways, indicating that either conversations about the terrorist event were undesirable or that such conversations were entirely acceptable. Each employee could then choose for themselves. For other disagreements, such as who deserves the most care, it can be difficult to find solutions that suit everyone’s needs.

Both leaders and staff should be aware of the importance of unity and peer support. Here, too, the foundation for success in times of crisis seems to be in the past, in that well-functioning work environments are more easily united, are more caring, and offer greater resilience to trauma. Thus, arguments for supportive management and a caring work environment should be supported in disaster planning.

## 6. Conclusions

The study provides information on specific issues at work and in the working environment that are important for employees in the aftermath of a workplace disaster. When researching the workplace in future disasters, questionnaires should, to a greater extent than traditional work environment surveys, emphasize conditions that are in line with these findings. Important factors to clarify are leaders’ ability to accept individual needs and help people cope with stress, the experience of being part of a community and working towards common goals, how employees take care of each other, individual and collective perceptions of work as a positive coping strategy, and prioritization and adaptation of work requirements.

## Figures and Tables

**Table 1 ijerph-18-01920-t001:** Characteristics of the participants in groups of exposure and PTSD.

Variables	Directly Exposed PTSD Group (*n* = 24)	Indirectly Exposed PTSD Group (*n* = 25)	Directly Exposed Non-PTSD Group (*n* = 24)	Indirectly Exposed Non-PTSD Group (*n* = 25)
Age, years	52 (10.1)	52 (11.5)	51 (9.9)	51 (10.4)
Gender				
Women	18	19	19	19
Men	6	6	5	6
Education, years	7.9 (2.3)	7.7 (2.9)	8.3 (3.0)	7.9 (3.6)
Doctor certified sick leave *	18	10	7	3

Data are given as mean (SD) or *n*. * In the four years following the terror attack.

## Data Availability

According to the approval from the Norwegian Regional committees for medical and health research ethics, the data are to be stored properly and in line with the Norwegian Law of privacy protection. Public availability would compromise privacy of the respondents. However, anonymized data are available to interested researchers upon request, pending ethical approval from our Ethics committee. Interested researchers can contact project leader Trond Heir (trond.heir@medisin.uio.no) with requests for the data underlying our findings.

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
