# Peer review of "The Role of Workplace on Work Participation and Sick Leave after a Terrorist Attack: A Qualitative Study"

_ijerph, 2021, doi:10.3390/ijerph18041920_

Round 1
Reviewer 1 Report
PTSD is an important issue of mental health. Especially the use of terrorist attack as the research object of PTSD is academically rare. This phenomenon shows the research importance of this article. However, based on academic requirements, I made the following review comments.
- In citing literature, PTSD has always been an important research topic for mental health. However, the authors cited a relatively long time in the literature, and should increase the citations of the literature in the last three years.
- During the in-depth interview, the authors divided the interviewees into directly exposed PTSD group, indirectly exposed PTSD group, directly exposed non-PTSD group, and indirectly exposed non-PTSD group. Obviously, the authors believe that these four types of interviewees have different psychological responses to terrorist attacks. In fact, the PTSD of the above four types of interviewees should indeed be different. However, the authors mixed the four types of interviewees in the results and discussion sections. I think this is inappropriate. It is recommended that the authors conduct in-depth discussions on the above four types of interviewees.
- In line 16 and 80, "2519" should be expressed as "2,519".
- In line 397, “Supportivemanagement” should be expressed as “Supportive management”.
- In line 426, “Senseofcohesion” should be expressed as “Sense of cohesion”.
- In line 454, “Workingasacopingstrategy” should be expressed as “Working as a coping strategy”.
- In line 487, “Highdemandsandlackofacceptance” should be expressed as “High demands and lack of acceptance”.
Author Response
Reviewer 1
Thank you for your responses, of which we have followed most of the recommendations.
1. In citing literature, PTSD has always been an important research topic for mental health. However, the authors cited a relatively long time in the literature, and should increase the citations of the literature in the last three years.
Reply; We have replaced two references in the introduction to be more up to date (reference 2 and 3). Should there be specific papers beyond this that should be cited, we are happy for suggestions.
2. During the in-depth interview, the authors divided the interviewees into directly exposed PTSD group, indirectly exposed PTSD group, directly exposed non-PTSD group, and indirectly exposed non-PTSD group. Obviously, the authors believe that these four types of interviewees have different psychological responses to terrorist attacks. In fact, the PTSD of the above four types of interviewees should indeed be different. However, the authors mixed the four types of interviewees in the results and discussion sections. I think this is inappropriate. It is recommended that the authors conduct in-depth discussions on the above four types of interviewees.
Reply: We are sorry for what may be a misunderstanding. The division was made in the sampling procedure to provide a wide range of participants with different experiences.In the results section, we have indicated how many of those who stated a particular theme were directly exposed and how many had had PTSD, thus showing that the themes that were highlighted were relatively independent of the degree of exposure or symptom burden. A more in-depth analysis of the importance of exposure or symptom level, we believe will require a quantitative approach.
3. In line 16 and 80, "2519" should be expressed as "2,519".
4. In line 397, “Supportivemanagement” should be expressed as “Supportive management”.
5. In line 426, “Senseofcohesion” should be expressed as “Sense of cohesion”.
6. In line 454, “Workingasacopingstrategy” should be expressed as “Working as a coping strategy”.
7. In line 487, “Highdemandsandlackofacceptance” should be expressed as “High demands and lack of acceptance”.
Reply: Thank you, this has been corrected
Reviewer 2 Report
Using qualitative methods, this study investigated factors in workplaces that are associated with government employees’ sick leave after a terrorist attack. The topic is interesting. I have a couple of comments for improvement
Title
- It might be more exact to be The Role of Workplace on work participation and sick leave after a terrorist attack: a qualitative research
Introduction:
- The introduction may be shortened. For example, the first two paragraphs may combine, where you just need to stress that work participation and sick leave after a terrorist attack is an important issue for research, and the role of work environment warrant attention. The third and the fourth paragraph may be combined, where you need to first gave us a literature review on what factors are reported to be associated with sick leave. Currently you just give a very brief sentence saying that “systematic reviews have stated that…”. Please provide more information on the methods that were used in previous research. Is there any research done using qualitative in-depth interview? As you said, this is the literature gap. But intuition tells me that this kind of work is not difficult conduct. Why no qualitative study has been done for this aim? Please make it clear to your readers.
- In line 64-66, authors pointed out the limitation that “The few studies that have investigated work-related factors and sick leave in the aftermath of traumatic events have used questionnaires that were developed for working life research in general”. The questionnaires should be concisely introduced to support your opinion.
materials and methods.
- The 100 participants were selected from 2519 employees who had completed a web-based three-wave survey after the terrorist attack. How were these 2519 employees chosen? Just by filling web-based three-wave survey basing on their own willingness? This section should be clarified.
- Please justify the samples size of 25 for each sub group.
- table 1, why you have triple numbers of women participants than men? How will this affect your findings?
- page 3 procedure. You mentioned that in the interview that you asked whether the participants had any sick leave after the attack, can you please tabulate out these findings in table 1? If there were so few participants experienced sick leave, how could you investigate the reason why they did it?
- methods section, please say more on how your semi-structure questionnaire, i.e. which particular question, could inform the respondents to report work environment factors that may affect sick leave. This is very important because it is your aim of study.
- In line 120-122, authors said that “The two authors responsible for the analysis (EHS and KPT) had not previously conducted work environment research, and they approached the data material with a minimum of their own interests or theories.” Is this meant that the analysis is objective or something other? This should be clarified.
Results:
- results may be improved in several ways. First, as you mentioned earlier, you have four groups of participants, they may have different experiences and responses to the work environment factors that were affecting sick leave, can you please specify? Second, men and women may have different responses to work environment, can you please specify? Third, the most important factor, people who experienced sick leave are expected to have very different feeling of the environment.
- Can you specify the reason why these participants had experienced sick leave? This direct reason might be more important than work environment factors. Saying, someone may catch a flue?
- The title of section 3.4 is “High demand and lack of acceptance”. But if we don’t reding the following paragraphs, it is hard to understand what is in high demand and what is lacking of acceptance. The title should be clear.
Discussion:
- many interpretations in the discussion section are groundless. I would suggest the authors to combine the results section and the discussion to tell a more coherent story.
- The second paragraph of section 4.2 discussed about the how the good cohesion work, but did not discussed how these characteristics of good cohesion worked in work participation.
- In line 522-524, two researchers performed the analysis together or separately, in order to guarantee the validity?
Author Response
Reviewer 2
Thank you for your extensive work in reviewing the article. We have taken note of most of the recommendations and incorporated them in the revised version. Sorry that the numbering of your points was lost during the copy-past procedure
Title
1. It might be more exact to be The Role of Workplace on work participation and sick leave after a terrorist attack: a qualitative research
Reply: Thanks. We have changed the title according to the proposal
Introduction:
2. The introduction may be shortened. For example, the first two paragraphs may combine, where you just need to stress that work participation and sick leave after a terrorist attack is an important issue for research, and the role of work environment warrant attention. The third and the fourth paragraph may be combined, where you need to first gave us a literature review on what factors are reported to be associated with sick leave. Currently you just give a very brief sentence saying that “systematic reviews have stated that…”. Please provide more information on the methods that were used in previous research. Is there any research done using qualitative in-depth interview? As you said, this is the literature gap. But intuition tells me that this kind of work is not difficult conduct. Why no qualitative study has been done for this aim? Please make it clear to your readers.
Reply: We have shortened the introduction section as recommended by combining the first two paragraphs. In the next paragraph we briefly introduce the knowledge about work environment and sick leave under normally stable conditions. We have sated this more clearly. We then summarize in more detail the sparse knowledge about associations between working environment and sick leave in the aftermath of traumatic events and specify that there is a lack of in-depth interviews.
3. In line 64-66, authors pointed out the limitation that “The few studies that have investigated work-related factors and sick leave in the aftermath of traumatic events have used questionnaires that were developed for working life research in general”. The questionnaires should be concisely introduced to support your opinion.
Reply: We have added information about the questionnaire used (line 55)
materials and methods.
4. The 100 participants were selected from 2519 employees who had completed a web-based three-wave survey after the terrorist attack. How were these 2519 employees chosen? Just by filling web-based three-wave survey basing on their own willingness? This section should be clarified.
Reply: We have added information that the 2519 individuals participated on a voluntary basis following a request to all employees in 14 of 17 Norwegian governmental ministries (line 72).
5. Please justify the samples size of 25 for each sub group.
Reply: We have stated that the high number of participants was chosen to ensure experience across departments, subdivisions, positions, and education (line 83).
6. table 1, why you have triple numbers of women participants than men? How will this affect your findings?
Reply: We have added a paragraph in the limitation section, addressing the higher proportion of female participants and the possibility of a gender bias of unknown direction.
7. page 3 procedure. You mentioned that in the interview that you asked whether the participants had any sick leave after the attack, can you please tabulate out these findings in table 1? If there were so few participants experienced sick leave, how could you investigate the reason why they did it?
Reply: Table 1 shows doctor-certified sick leave, which we have now stated more clearly (line 90). Due to the exceptional incident, it was possible for long periods to take a few days off without approved sick leave. Thus, many employees have been in the gray zone between full work participation and doctor-certified sick leave, which has given them experience in what kind of work-related factors contributed in one direction or another.
8. methods section, please say more on how your semi-structure questionnaire, i.e. which particular question, could inform the respondents to report work environment factors that may affect sick leave. This is very important because it is your aim of study.
Reply: We have added work-related to the factors in question (line 104).
9. In line 120-122, authors said that “The two authors responsible for the analysis (EHS and KPT) had not previously conducted work environment research, and they approached the data material with a minimum of their own interests or theories.” Is this meant that the analysis is objective or something other? This should be clarified.
Reply: We have added to the sentence about interests or theories: “that could have influenced the results or the choice of quotes in a desired direction” (line 115).
Results:
10. results may be improved in several ways. First, as you mentioned earlier, you have four groups of participants, they may have different experiences and responses to the work environment factors that were affecting sick leave, can you please specify? Second, men and women may have different responses to work environment, can you please specify? Third, the most important factor, people who experienced sick leave are expected to have very different feeling of the environment.
Reply: The stratified recruitment procedure was chosen to ensure that the study included a wide range of participants in terms of exposure and stress reactions. We did not intend to compare subgroups, neither in terms of exposure, stress level, gender or whether they had sick leave. We were looking for the broadest possible experience base for what was perceived as important for being at work in the turbulent times after a workplace disaster. In fact, there was no previous evidence of this, and there existed no questionnaires with specific work environment questions related to disaster management.
In the results section, we have indicated how many of those who stated a particular theme were directly exposed and how many had had PTSD, thus showing that the themes that were highlighted were relatively independent of the degree of exposure or symptom burden. A more in-depth analysis of subgroups due to exposure, symptom level, gender, or reasons for absence, we believe will require a quantitative approach.
11. Can you specify the reason why these participants had experienced sick leave? This direct reason might be more important than work environment factors. Saying, someone may catch a flue?
Reply: We have previously published findings of the total sample that for those directly exposed, doctor-certified sick leave in the years after the Oslo bombing was twice as high as the sick leave prior to the attack (ref.no.5). The increase in doctor-diagnosed diagnoses did not occur in specific disease groups. Stress appeared to result in sick leave with unexpected diagnoses. This type of reasoning is difficult enough to do on large samples, and impossible to do on such small samples as in the present study. In addition, we were not allowed by the ethics committee to state the reason for sick leave in small groups, because these were vulnerable data that could compromise privacy of the participants.
12. The title of section 3.4 is “High demand and lack of acceptance”. But if we don’t reding the following paragraphs, it is hard to understand what is in high demand and what is lacking of acceptance. The title should be clear.
Reply: We have changed the title of section 3.4 to “High work demands and lack of acceptance for individual needs” (line 383).
Discussion:
13. many interpretations in the discussion section are groundless. I would suggest the authors to combine the results section and the discussion to tell a more coherent story.
Reply: We have combined the results section and the discussion by discussing each theme after it is presented in the results section.
14. The second paragraph of section 4.2 discussed about the how the good cohesion work, but did not discussed how these characteristics of good cohesion worked in work participation.
Reply: We agree, the relationship between cohesion and work participation is discussed in the first paragraph. We believe that the second paragraph is still useful to include, because promoting cohesion was a key issue, and because the collective responsibility for this is important to convey for disaster management in the future.
15. In line 522-524, two researchers performed the analysis together or separately, in order to guarantee the validity?
Analyses were performed separately. This is included (line509)
Reviewer 3 Report
Dear Authors,
I congratulate you for the very interesting and well written manuscript on this important topic on how to recuperate people to their work lives and lives in general through work after a terrorist attack.
Besides the topic, the sample is quite large for a qualitative study, and the results are presented in a clear and well organised structure.
I just offer you a few comments you might consider to improve the paper:
- Introduction. Sometimes I got confused with the concept "work participation" as for instance in page 2 lines 73-74 it seems you are using three categories: work participation, sick leave or returning to work. Maybe you could clarify it, my guess is that by work participation means workers came on day 2 back to work, but it is not clear to me.
- In p2, line 56 you state reorganizational stress, maybe you meant organizational stress.
- Table 1. Please explain in the text "Sick leave after 4 years", I understand you asked them in the interview whether they are still on sickleave after the attack. Is that right? Also, I feel the results focus more on people at work, or returning to work, and that the comments on the still on sick leave are more reduced. I miss some information on them. It would be interesting to add a paragraph somewhere adding relevant information on them which you may have gathered during the interviews (did they look for professional help? are they terrified to come back to the building? did they recover their lives and work somewhere else?), maybe they offered some arguments besides lack of attention to individual needs or high demands. If not you could point it out as a limitation or future research.
- I miss information on the managers. You mention the good managers prior to the attack may have behaved as better managers after the attack, I find this interesting. But nowhere do you mention whether the managers were impacted by the attack (most likely too), whether they received any training to or briefing to give support, relax on objectives, communicate with coworkers, etc. THis is another limitation to add to the study, and future research, how the degree or type of impact on managers affect their support to their teams or departments.
- I think an important point that was coming to my head throughtout the reading is how much of these benefits of work depend on what managers or the ministry does, and how much on how the recipients receive the same policies. You mention it in page 11 lines 501-503, but I think this should be a general comment in the discussion rather than inserted in section 4.4. Same for lines 510-514 in page 12.
- I understand that the two researchers analysing the data went on the task with an open mind not biased by theories. But maybe in the discussion you could add some theoretical contribution by an overall explanation of your results by supporting them with theories related to stress, demands, control, individual differences, social support, etc.
- I miss a section on Limitations and another on Future research. WHere do we go from here? what do you suggest other researchers should do better if facing a similar sample? or how does this inspire changes in standard questionnaires to adapt them to this type of situations.
Overall, congratulations. I enjoyed reading your research!
Author Response
Reviewer 3
Thanks for the kind words and many good suggestions, of which we have incorporated some in the article. Sorry that the numbering of your points was lost during the copy-past procedure.
1. Introduction. Sometimes I got confused with the concept "work participation" as for instance in page 2 lines 73-74 it seems you are using three categories: work participation, sick leave or returning to work. Maybe you could clarify it, my guess is that by work participation means workers came on day 2 back to work, but it is not clear to me.
Reply: We acknowledge the ambiguities. The employees had several absences, some doctor-certified sick leave, some absences as an agreement when it was difficult to be at work. We wanted to investigate what contributed to absence from work and what contributed to people staying at work. To make the aim clearer, we have added ‘returning to work after absenteeism’.
2. n p2, line 56 you state reorganizational stress, maybe you meant organizational stress.
Reply: Yes, thank you. It has been corrected.
3. Table 1. Please explain in the text "Sick leave after 4 years", I understand you asked them in the interview whether they are still on sickleave after the attack. Is that right? Also, I feel the results focus more on people at work, or returning to work, and that the comments on the still on sick leave are more reduced. I miss some information on them. It would be interesting to add a paragraph somewhere adding relevant information on them which you may have gathered during the interviews (did they look for professional help? are they terrified to come back to the building? did they recover their lives and work somewhere else?), maybe they offered some arguments besides lack of attention to individual needs or high demands. If not you could point it out as a limitation or future research.
Reply: We asked about sick leave in the 4 years following the attack. We did not focus on people still on sick leave.
4. I miss information on the managers. You mention the good managers prior to the attack may have behaved as better managers after the attack, I find this interesting. But nowhere do you mention whether the managers were impacted by the attack (most likely too), whether they received any training to or briefing to give support, relax on objectives, communicate with coworkers, etc. THis is another limitation to add to the study, and future research, how the degree or type of impact on managers affect their support to their teams or departments.
Reply: We have added this to the limitations (line 516).
5. I think an important point that was coming to my head throughtout the reading is how much of these benefits of work depend on what managers or the ministry does, and how much on how the recipients receive the same policies. You mention it in page 11 lines 501-503, but I think this should be a general comment in the discussion rather than inserted in section 4.4. Same for lines 510-514 in page 12.
Reply: We absolutely agree that this is a key point, but we are unsure of how and where to emphasize it further. We believe it comes out well in the results and have little desire just to repeat it in the discussion.
6. I understand that the two researchers analysing the data went on the task with an open mind not biased by theories. But maybe in the discussion you could add some theoretical contribution by an overall explanation of your results by supporting them with theories related to stress, demands, control, individual differences, social support, etc.
Reply: We have considered expanding the discussion as recommended. However, we primarily prefer to keep the discussion within the current framework, considering the length of the whole paper. The findings are extensive and are suitable for many theories of which we have limited the discussion to some.
7. I miss a section on Limitations and another on Future research. WHere do we go from here? what do you suggest other researchers should do better if facing a similar sample? or how does this inspire changes in standard questionnaires to adapt them to this type of situations.
Reply: We have renamed a section to Strengths and limitations (line 501) and added some more limitations. We have also added a paragraph with research implications (line 557).
Reviewer 4 Report
the manuscript entitled "Work participation or sick leave after a terrorist attack: The role of the workplace" aims to identify work-related factors that are perceived as important for work participation versus sick leave after a terrorist attack.
The paper highlights a very delicate issue for the health of the worker that concerns a specific situation: recovering from an attack in the workplace and the consequences that this has brought over time. However, the article is a bit dated and concerns a sample hit in 2011 and interviewed in 2015. 5 years have passed and today it would have been important to check the health of the workers. Stratified sampling is adequate but precisely because there are workers suffering from Post Traumatic Stress Syndrome it would have been important to verify today the persistence of the malaise.
Furthermore, the qualitative methodology used is insufficient, they could instead have revised the individual pieces of the interview, detected the headwords and through the K means method, an analysis of the clusters that would have allowed them a greater synthesis and therefore possible explanation of the phenomenon.
For these reasons I do not consider the paper suitable for the journal
Author Response
Reviewer 4
the manuscript entitled "Work participation or sick leave after a terrorist attack: The role of the workplace" aims to identify work-related factors that are perceived as important for work participation versus sick leave after a terrorist attack.
The paper highlights a very delicate issue for the health of the worker that concerns a specific situation: recovering from an attack in the workplace and the consequences that this has brought over time.
However, the article is a bit dated and concerns a sample hit in 2011 and interviewed in 2015. 5 years have passed and today it would have been important to check the health of the workers.
Reply: We regret that it has taken a long time to publish our data. However, terrorism is still relevant, and in our opinion the data are still interesting today. No similar investigations have been published either before or after this terror attack. As for the health of the employees, we followed this up for three years after the incident, which has been published. We agree that it could be interesting with a follow-up after 10 years and will consider the proposal.
Stratified sampling is adequate but precisely because there are workers suffering from Post Traumatic Stress Syndrome it would have been important to verify today the persistence of the malaise.
Reply: We agree. However, this is not in contrast to the need to publish data from the incident, which has implications for future research and for future planning and management of similar disasters.
Furthermore, the qualitative methodology used is insufficient, they could instead have revised the individual pieces of the interview, detected the headwords and through the K means method, an analysis of the clusters that would have allowed them a greater synthesis and therefore possible explanation of the phenomenon.
Reply: We recognize that there may be methodological approaches that are as good or perhaps even better than our choice. However, the chosen method is widely used and recognized in qualitative research.
For these reasons I do not consider the paper suitable for the journal
Round 2
Reviewer 2 Report
no further comments
Author Response
Reviewer 2 has ticked the box “Moderate English changes required”.
The first manuscript that we submitted was edited linguistically by San Francisco Edit. Following the reviewers' recommendation, we have also had our revised manuscript edited by the same company, so that all changes in the revised version should be linguistically correct.
Otherwise, the reviewer states: "No further comments"
Reviewer 4 Report
The authors do not bring any changes and add nothing with respect to the reflections that I have given them, even if they share them. I, therefore, remain in my previous position. The paper as it is presented seems more related to a research report whose data if presented so dated should at least have a follow-up (which however the authors are willing to do later).
In my opinion for the level of the journal, the study has great limits.
Author Response
We recognize that the reviewer’s assessments moved from "Must be improved" on all assessment points to "Can be approved" on all but one point, after the revision of our manuscript. The remaining point seems to have to do with the reviewer’s view that the study should at least have a follow-up.
We think that these results should be presented for themselves, and that a simultaneous publication of a follow-up study will be too extensive for one and the same publication. The current paper is already comprehensive in itself.
Our research group has published several papers about the terrorist attack on the Norwegian ministries, and some of them have been published in high-impact journals. From our point of view, with considerable experience in disaster psychiatry and management, the present study is perhaps the most important of them all. It has been very resource-intensive for both participants and researchers, and it is based on recognized research methods. We are proud of the implications for both practical purposes and future research. We believe the study will get the attention of many readers, and we really hope that the reviewer will acknowledge this.
Finally, we are grateful for the interest in a follow-up study.